# Determinants of maternal near miss among women admitted to maternity wards of tertiary hospitals in Southern Ethiopia, 2020: A hospital-based case-control study

**Aklilu Habte**[1]*, **Merertu Wondimu**[2]

**1** Department of Public Health, College of Medicine and Health Sciences, Wachemo University, Hosaena, Southern Ethiopia, **2** School of Nursing and Midwifery, Faculty of Health Science, Institute of Health, Jimma University, Jimma, Southwest Ethiopia

* akliluhabte57@gmail.com

## Abstract

### Background

A maternal near-miss (MNM) refers to when a gravely ill woman survives a complication as a result of the standard of care she receives or by chance during gestation, childbirth, or within 42 days of the termination of pregnancy. Rescuers of near-miss events share several features with mothers who have died and identifying MNM determinants will aid in improving the capacity of the health system to reduce severe maternal morbidity and mortality. Ethiopia is one of the countries in sub-Saharan Africa with high maternal mortality and morbidity, but there is little evidence on determinants of a MNM based on a WHO criteria. Hence, this study aimed at identifying determinants of MNM among women admitted to tertiary hospitals in southern Ethiopia, 2020.

### Methods

A facilities-based unmatched case-control study was conducted in five selected tertiary hospitals found in central southern Ethiopia from February 1 to June 1, 2020. A total of 322 (81 cases and 241 controls) study participants were included in the study. At the time of their discharge, cases were recruited consecutively, while controls were selected using a systematic sampling method. The cases were women admitted to hospitals during pregnancy, childbirth, or 42 days following termination of pregnancy who met at least one of the WHO near-miss criteria. Whereas the controls comprised of women who were admitted during pregnancy, childbirth, or 42 days following termination of pregnancy and discharged without severe obstetric complications. Data collection was conducted using the interviewer-administered structured questionnaire and data abstraction tool. The data was coded and entered into Epi-Data version 3.1 and exported to SPSS version 23 for analysis. Multivariable logistic regression analysis was conducted and determinants of MNM were established at p-value<0.05.

**Data Availability Statement:** All relevant data are within the paper and its Supporting Information files.

**Funding:** The author(s) received no specific funding for this work.

**Competing interests:** The authors have declared that no competing interests exist

**Abbreviations:** ANC, Antenatal Care; AOR, Adjusted Odds Ratio; BPCR, Birth Preparedness and Complication Readiness; CS, Caesarean section; EDHS, Ethiopia Demographic and Health Survey; MMR, Maternal Mortality Ratio; MNM, Maternal Near Miss; WHO, World Health Organization.

## Results

Severe postpartum hemorrhage (50.6%) and sepsis (23.4%) were the most common reasons for admission of cases. Lack of ANC [AOR = 3.25; 95%CI: 2.21,7.69], prior history of Cesarean section [AOR = 3.53; 95%CI:1.79,6.98], delaying more than 60 minute to access final place of care [AOR = 3.21; 95%CI:1.61,6.39], poor practice of Birth preparedness and complication readiness (BPCR) [AOR = 3.31; 95%CI:1.50,7.29], and history of preexisting medical disorders [AOR = 2.79; 95%CI:1.45,5.37] were identified as significantly determinants of maternal near miss.

## Conclusion and recommendation

Stakeholders need to enhance their efforts for improving access to roads and transportations. Besides, women who have a prior history of Caesarean section, chronic medical conditions, and no ANC need special attention from their families and health care providers to proactively mitigate the occurrence of serious obstetric complications. More attention has to be paid to birth preparedness and complication readiness activities.

## Introduction

According to UN inter-agency figures, the global maternal mortality ratio decreased by 38% from 2000 to 2017, from 342 deaths to 211 deaths per 100,000 live births with a reduction of lifetime risk of maternal death from 1 in 100 to 1 in 190 [1]. Sub-Saharan Africans suffer 533 maternal deaths per 100,000 live births, or 200,000 maternal deaths annually, representing the highest maternal mortality ratio [1, 2]. Ethiopia is one of five countries that account for half the maternal deaths worldwide [2]. Since 2000, Ethiopia has cut maternal mortality by more than half, but the maternal mortality ratio is still too high, with 401 per 100,000 live births accounting for nearly 14,000 maternal deaths [3].

The most devastating end to a pregnant woman is maternal death, and it is sometimes characterized as only the "tip of the iceberg," while maternal morbidity is the "base," and many more will live for every woman who dies, but often suffer from lifelong disabilities [4]. The world health organization (WHO) working groups describe maternal near-miss (MNM) as a woman who nearly died but survived a complication that occurred during pregnancy, childbirth, or within 42 days of termination of pregnancy, due to the quality of care she receives or by chance [5, 6]. For MNM cases, the diagnostic criteria are potentially life-threatening pregnancy conditions; the provision of critical interventions; and organ dysfunction [5, 7, 8].

There is a greater occurrence rate of MNM than maternal mortality. Studies conducted elsewhere have shown that MNM is 15–26 times more frequent in low-resource settings than maternal death [6, 9–11]. Near miss is a multifactorial condition triggered by socio-economic, health events, health care provider competence, and sub-standardized facilities [12–14]. Studies have shown that predictors of maternal near misses are, previous cesarean section, pre-existing medical disorders, induction of labor, and lack of antenatal care [15, 16]. Delays at home, on the way to health facilities, and at the health facility, themselves play an important role in raising the size of MNM [12, 17–19]. Studies conducted in Brazil and Gabon found that delays were linked with 68.7% and 40% of near misses [20, 21], respectively. All these delays could be minimized by setting a Birth preparedness and complication readiness(BPCR) plan [22].

Even though the study of maternal mortality is increasingly difficult, the maternal mortality ratio (MMR) is the key measure of the standard of obstetric care [23–25]. Compared to the MMR, a review of MNM cases is quite important for assessing what goes wrong in pregnancy-related care when an MNM happens in a specific environment [26, 27]. It highlights the quality of obstetric care at a health facility and offers useful information to track factors contributing to maternal death by clearly notifying those challenges that had to be addressed [5, 17, 28, 29].

Despite all the efforts on maternal health care, maternal near-miss, disabilities, and deaths were exceptionally high in developing countries, including Ethiopia [30]. The WHO Technical Working Group points out that the MNM strategy should have been included in national policies to improve maternal health [5]. The adoption of the MNM approach in the healthcare system would help to determine the occurrence of severe maternal complications and deaths, evaluate the health system's performance in reducing severe maternal outcomes, and recognize the use of key interventions to prevent and reduce serious complications [5].

Ethiopia is one of the countries in sub-Saharan Africa with high maternal mortality and morbidity, but there is little evidence of risk factors for a near miss. Hence this study aimed to identify determinants of maternal near-miss among women admitted in tertiary hospitals in southern Ethiopia. Unlike to previous studies [9, 16], which focused only on potentially life-threatening conditions and overlooked the clinical criteria like organ dysfunction and clinical intervention provision, the current study was conducted in fully equipped hospitals and focused on all three WHO criteria [5]. This might make the evaluation criteria more stringent and show a better picture of near-miss events. Consequently, this research may be used as an input for comprehensive interventions to strengthen district health services by tracking the quality of care, assessing the implementation of key interventions, and alerting referral processes.

## Methods and material

### Study setting, period, and design

A facility-based unmatched case-control study was conducted in five selected Tertiary hospitals found in Southern Ethiopia from February 1 to June 1, 2020. Hawassa, the administrative center of the region, is located 279 km from Addis Ababa (the capital city of Ethiopia). In the region, there are 12 Tertiary hospitals, all of which have well-organized laboratory and surgical facilities for comprehensive emergency obstetric care, with over 124,000 obstetric cases treated each year, including over 4000 near miss cases. All of the hospitals were designed in the way to manage maternal near-misses events.

### The population of the study

The source population for cases was all mothers who were admitted to maternity wards of hospitals in southern Ethiopia during pregnancy, childbirth, or the first 42 days after giving birth and nearly died due to complications but survived. Selected mothers from selected hospitals who met at least one of WHO near-miss criteria [5] were taken as study populations for cases. The three criteria were serious maternal complications related to pregnancy and childbirth, provision of critical care, and organ dysfunction. All mothers admitted to tertiary hospitals in southern Ethiopia during pregnancy, childbirth, or within the first 42 days after giving birth, and who did not have any of the complications mentioned in the WHO near-miss criteria, were considered as a source population for control. The study population for controls consisted of those selected mothers from the maternity wards of the selected hospital and discharged without any of the above-mentioned complications. Those mothers who were initially

chosen as control and discharged but unfortunately returned as a case and those mothers who were seriously ill during the time of data collection were excluded from the study.

## Sample size determination

By applying a double population proportion formula via the stat calc menu of Epi Info 7 software for un-matched case-control, the sample size for the study was determined. The sample size was estimated from various literature for the key explanatory variables associated with MNM and then the variable resulted in a large sample size was taken. During the estimation of the sample size, the following assumptions were considered: 95% confidence level, 80% power, the ratio of the case to control 1:3, percent of controls exposed 40.5%, and percent of cases exposed 60.2%. The percent of cases and controls were taken from the study conducted in northern Ethiopia that delays in reaching the final place of care by more than 60 minutes were the most determinant factor for the maternal near-miss [16]. Based on the above assumptions the estimated sample size was 292 (73 cases and 219 controls). After considering the nonresponse of 10%, the final sample size used for this study was 322(81 cases and 241 controls).

**Sampling procedures.** Out of the 13 tertiary hospitals in southern Ethiopia, five hospitals were selected by lottery method: Worabe tertiary hospital, Butajira tertiary hospital, Attat Tertiary hospital, Nigist Eleni Mohammed Memorial tertiary hospital, and Mercy tertiary hospital. Obstetric case management reports and registration books were observed over the last three months (February 1-May 1) to assess the obstetric client/patient flow rate of the respective hospitals. The sample size for each hospital was proportionally distributed based on their client flow as follows: Worabe tertiary hospital (21 cases and 62 controls), Butajira tertiary hospital (25 cases and 75 controls), Attat Tertiary hospital (9 cases and 26 controls), Nigist Eleni Mohammed Memorial tertiary hospital (14 cases and 42 controls), and Mercy tertiary hospital (12 cases and 36 controls). Cases were then recruited consecutively at the time of the discharge, while controls were picked by using a systematic method of sampling. The interval (K) was determined by dividing the average number of controls who visited each hospital in the previous three months by the proportionally allocated control sample size for each hospital.

## Data collection tool, procedure, and personnel

The data were collected using a pre-tested, standardized, interviewer-administered questionnaire, along with a standardized Near-miss abstraction checklist after possible adjustment to the local context [5, 7, 16, 25, 26, 31]. The WHO near-miss assessment criteria have been used to classify cases and control [5]. The questionnaire consisted of five main parts: socio-economic and demographic variables of mothers, obstetric characteristics, medical history, practice towards birth preparedness and complication readiness (BPCR) plan, and maternal health service-related characteristics. A tool adapted from the EDHS 2016 report was used to assess the socioeconomic status of study participants' households [32]. Data that could not be accessed by interviews such as obstetric complications diagnosis, management provided, and findings of laboratory investigation were extracted from patient medical records and discharge summaries. As data collectors, five midwife nurses who have obstetric care experience (one per hospital) and who can speak the local language were recruited. As supervisors, five public health professionals who have a bachelor's degree have been recruited.

## Data quality management

To ensure accuracy, the questionnaire was prepared in English, translated to Amharic by experts in that language, and back-translated to English separately by two individuals. Both data collectors and supervisors were provided with intensive training lasting one day which

aimed at data collection procedure, the objective of the study, the contents of the question-naires, how to approach the respondents, how to deal with the difficulties that might occur during the data collection process, and on the issues of confidentiality and privacy. One week before the actual data collection, a pre-test was conducted at Wolliso specialized hospital for 5 percent of the sample size (4 cases and 12 controls), and feedback was incorporated accordingly to better completion of the questionnaires. All healthcare workers operating in each hospital's MNCH case team have been sensitized to the issue to alert enumerators when they suspect near-miss events. Besides, on the wall of each ward, the parameters for the MNM were posted. Frequent checks were carried out by the Principal Investigator and Supervisors on the consistency and completeness of the data gathered and appropriate corrections were made on the spot.

## Definition and operationalization of variables

MNM case: The MNM case was taken into account when the admitted mother faced at least one of the WHO criteria, but survived: 1)Significant pregnancy-related maternal complications (e.g. Severe postpartum hemorrhage, Severe preeclampsia, Eclampsia, Sepsis or severe systemic infection, or Ruptured uterus); 2) Provision of critical care (any blood transfusion, laparotomy, or admission to intensive care unit), and 3)Organ dysfunction (cardiovascular dysfunction, Respiratory dysfunction, Renal dysfunction, Hepatic dysfunction, Neurological dysfunction, Uterine dysfunction, and coagulation/hematological dysfunction [5]. All of these evaluations were undertaken by either general practitioners, gynecologists, and the data were taken from, medical records of mothers by data collectors.

The first maternal delay: was the period between identification of health problems and decision-making to pursue maternal health care. A delay was deemed to take more than 24 hours to decide to seek treatment, otherwise no delay [16].

Second maternal delay: was a time after decision-making to reach health facilities. The time has been estimated at more than one hour to reach the existing health facility and otherwise not [16, 33].

Third maternal delay: was the interval of time between reaching the health facility and accessing the services needed. It took more than 1 hour to receive a delivery service deemed delay and less than an hour deemed no delay [33].

Good Birth Preparedness and complication readiness: Described as having taken at least five of the eight measures recommended by WHO: ascertained birthplace; identified birth attendants; put money aside; established emergency transportation; identified labor and birth companion; identified nearby health facility; identified blood donors if necessary, and identified care provider for children at home when the mother was away [34, 35].

Knowledge on key pregnancy danger signs: A woman was classified as knowledgeable if at least two of the four key signs of pregnancy (vaginal bleeding, severe headache, blurred vision, and swelling of the feet or face) were actively stated; if not, she was classified as not knowledgeable [36, 37].

Autonomy in household decision making: A woman was said to be autonomous in using MNCH services if she decided to receive MNCH services alone or with her husband (jointly); otherwise (if her husband decided alone or a third party) she was considered as non-autonomous [37, 38].

## Data analysis

The data was coded and entered into Epi-Data version 3.1 and exported to SPSS version 23 for analysis. For both cases and controls, univariate analyses such as frequency, proportion, mean

and standard deviation were computed. The wealth status of each household was examined using principal component analysis (PCA). Initially, 32 items were used, which were then categorized into 6 categories: household productive and non-productive assets, livestock ownership, crop production in quintals, approximate average monthly income, farmland in hectares, and residential homes with infrastructure. Since it would be difficult to differentiate between richer and poorer households if a variable/asset was owned by more than 95% or less than 5% of the sample, it was removed from the analysis. Following the first exploration of the variables using frequencies, the Kaiser-Meyer-Olkin measure of sampling adequacy (KMO>0.6) and Bartlett's test of sphericity (p-value<0.05) were used to determine if the assumptions for PCA have been met. Variables with anti-image correlations and commonalities less than 0.5, as well as variables with a loading (correlations greater than 0.4) in more than one component (having complex structure) and a single variable loading in a component, were excluded in each step until the iterations met the requirements. Finally, three components were derived from the PCA that explained a total variance of 71.7%, with the first component explaining the largest variance of 37.8%. Ownership of agricultural land, ownership of livestock, and household assets remained in the first component. To classify the household's wealth status, the component with the maximum variance was split into quintiles [32].

The Chi-square test was used to compare the proportion of cases and controls between selected categorical variables. To recognize the determinants of MNM, bivariable, and multivariable logistic regression analyses were used. In the bivariable analysis, explanatory variables with a p-value of <0.25 were simultaneously inputted into a multivariable logistic regression analysis model to monitor the influence of confounding variables. With their 95% confidence interval, crude and adjusted odds ratios were determined to assess the strength and existence of an association. MNM determinants were identified in the final model at p-value<0.05 and the Adjusted Odds Ratio (AOR) with a 95% CI. Finally, the results are summarized through texts, tables, and figures.

## Ethical approval and consent to participate

Ethical clearance was obtained from the Institutional Research Review Board(IRB) of Wachemo University, College of Medicine and Health Sciences. Written permission was obtained from the Southern Nation Nationality and People Regional Health Bureau, Zonal Health Departments, and participating hospitals. For those aged 18 and over, written informed consent was obtained from study participants after they were relieved of their health emergency (on discharge to home) and given details of the study objectives. Besides, consent was taken from a parent or guardian using normal disclosure processes for those participants less than 18 years of age. Participants' privacy and confidentiality were ensured prior to data collection. There was also voluntary participation in the study, and participants were told of the right to withdraw from the study at any time.

## Results

### Socio-demographic characteristic of respondents

A total of 322 participants (81 cases and 241 controls) were interviewed, yielding a 100% response rate. The mean age (±SD) for cases and controls was 29.67 (±4.64) and 28.92 (±4.24) years, respectively. On the Chi-square test, however, the mean age difference between cases and controls was not statistically significant. 77.9% and 85.5% of cases and controls belong to the 20–34 years age groups, respectively. Over half of the cases (50.6%) were rural residents, with the majority (61.4%) of the controls being urban residents. Regarding education status, there has been no formal education for 42(51.9%) cases and 82 (34.0%) controls. A high

proportion of cases (28.4%) are in the lowest quintile of wealth compared to controls (17.0%) (Table 1).

## Obstetric characteristics of respondents

Over one in ten mothers (12.7%) had their first child before the age of 16 years. The proportion of early pregnancies among near-miss groups was higher than in the control groups, 19.8%, and 10.4%, respectively. In comparison to 30.7% of controls, nearly two-fifth (38.2%) of cases had five or more pregnancies. There had been a comparable number of five or more children, 19.8% and 17.6 percent, for both cases and controls respectively. Likewise, the experience of prior abortion among cases and controls was 18.5% and 20.3%, respectively. Nearly half (49.4%) of cases and about one-fifth (19.1%) of controls had a history of cesarean delivery of at least one (Table 2).

**Table 1. Socio-demographic and economic characteristics of mothers admitted to tertiary hospitals in Southern Ethiopia, 2020.**

| Variable categories | Cases = 81 | Controls = 241 | Total = 322 | $X^2$ | P-value |
|---|---|---|---|---|---|
| | n (%) | n (%) | n (%) | | |
| **Age of mother in years** | | | | | |
| 35+ | 10(12.2) | 22(9.1) | 32(10.0) | 2.929 | 0.231 |
| 20–34 | 63(77.9) | 206(85.5) | 269(83.5) | | |
| <20 | 8(9.9) | 13(5.4) | 21(6.5) | | |
| **Residence** | | | | | |
| Urban | 40(49.4) | 148(61.4) | 188(48.4) | 3.61 | 0.057 |
| Rural | 41(50.6) | 93(38.6) | 134(41.6) | | |
| **Marital status** | | | | | |
| In marital union | 77(95.1) | 226(93.8) | 303(94.1) | 0.181 | 0.671 |
| Not in marital relation | 4(4.9) | 15(6.2) | 19(5.9) | | |
| **Religion** | | | | | |
| Orthodox | 22(27.2) | 89(36.9) | 111(34.5) | | |
| Muslim | 21(25.9) | 67(27.8) | 88(23.3) | | |
| Protestant | 37(45.7) | 75(31.1) | 112(34.8) | | |
| Catholic | 1(1.2) | 10(4.1) | 11(3.4) | | |
| **Mother's Educational level** | | | | | |
| No formal education | 42(51.9) | 82(34.0) | 124(38.5) | 9.978 | 0.030 |
| Primary education (1-8th) | 16(19.8) | 52(21.6) | 68(21.1) | | |
| Secondary(9-12th) | 13(16.0) | 63(26.1) | 76(23.6) | | |
| College and above | 10(12.3) | 44(18.2) | 54(16.8) | | |
| **Husband's Education** | | | | | |
| No formal education | 25(30.9) | 76(31.5) | 101(31.4) | 0.848 | 0.838 |
| Primary education (1-8th) | 29(35.8) | 74(30.7) | 103(31.9) | | |
| Secondary(9-12th) | 15(18.5) | 49(20.3) | 64(19.9) | | |
| College and above | 12(14.8) | 42(17.4) | 54(16.8) | | |
| **Wealth index** | | | | | |
| Highest | 13(16.0) | 51(21.2) | 64(19.8) | 3.552 | 0.470 |
| Fourth | 13(16.0) | 52(21.6) | 65(20.3) | | |
| Middle | 16(19.8) | 48(19.9) | 64(19.8) | | |
| Second | 19(23.5) | 46(19.1) | 65(20.3) | | |
| Lowest | 20(24.7) | 44(18.3) | 64(19.8) | | |
| Family size | | | | | |
| <5 | 37(45.7) | 125(51.9) | 162(50.3) | 0.929 | 0.335 |
| ≥5 | 44(54.3) | 116(48.1) | 160(49.7) | | |

**Table 2. Obstetric characteristics of mothers admitted to tertiary hospitals in Southern Ethiopia, 2020.**

| Variable categories | Cases = 81 | Controls = 221 | Total = 322 | $X^2$ | P-value |
|---|---|---|---|---|---|
| | n(%) | n(%) | n (%) | | |
| **GA during last delivery** | | | | | |
| Term | 47(58.0) | 223(92.6) | 270(83.9) | | |
| Preterm | 18(22.2) | 14(5.8) | 32(9.9) | | |
| Post-term | 16(19.8) | 4(1.6) | 20(6.2) | | |
| **Age at first pregnancy in years** | | | | | |
| ≥20 | 40(49.1) | 166(68.9) | 206(64.0) | 10.455 | 0.005 |
| 16–19 | 25(30.9) | 50(20.7) | 75(23.3) | | |
| <16 | 16(19.8) | 25(10.4) | 41(12.7) | | |
| **Gravidity** | | | | | |
| 1 | 8(9.9) | 28(11.6) | 36(11.2) | 1.595 | 0.451 |
| 2–4 | 42(51.9) | 139(57.7) | 181(56.2) | | |
| ≥5 | 31(38.2) | 74(30.7) | 105(32.6) | | |
| **Parity** | | | | | |
| 0(Nulliparous) | 11(13.6) | 41(17.0) | 52(16.1) | 1.698 | 0.637 |
| 1(Primiparous) | 13(16.0) | 29(12.0) | 42(13.1) | | |
| 2-4(Multiparous) | 41(50.6) | 131(54.4) | 172(53.4) | | |
| ≥5(Grand multiparous) | 16(19.8) | 40(17.6) | 56(17.4) | | |
| **Birth interval** | | | | | |
| ≥24 months | 38(46.9) | 163(67.6) | 201(62.4) | 11.097 | 0.001 |
| <24 months | 43(53.1) | 78(32.4) | 121(37.6) | | |
| **Desire on the last pregnancy** | | | | | |
| Planned | 48(59.3) | 169(70.1) | 217(67.4) | 2.57 | 0.071 |
| Not planned | 33(40.7) | 72(29.9) | 105(32.6) | | |
| **Last birth outcome** | | | | | |
| Live birth | 74(91.4) | 225(93.4) | 299(92.8) | 0.367 | 0.350 |
| Stillbirth | 7(8.6) | 16(6.6) | 23(7.2) | | |
| **Previous history of any obstetric complication/s@** | | | | | |
| Yes | 29(35.8) | 82(34.0) | 111(34.5) | 0.085 | 0.788 |
| No | 52(64.2) | 159(66.0) | 211(65.5) | | |
| **Ever had abortion** | | | | | |
| Yes | 15(18.5) | 49(20.3) | 64(19.9) | 0.125 | 0.872 |
| No | 66(81.5) | 192(79.7) | 258(80.1) | | |
| **Frequency of abortion** | | | | | |
| Once | 7(8.6) | 32(13.3) | 39(12.1) | | |
| More than once | 8(9.9) | 17(7.0) | 25(7.8) | | |
| **Previous history of C/S** | | | | | |
| Yes | 40(49.4) | 46(19.1) | 86(26.7) | 28.425 | <0.001 |
| No | 41(50.6) | 195(80.9) | 236(73.3) | | |
| **Occasions for C/S** | | | | | |
| Emergency | 31(38.3) | 39(16.2) | | | |
| Elective | 9(11.1) | 7(2.9) | | | |
| **Frequency of C/S** | | | | | |
| 1 | 21(25.9) | 32(13.3) | | | |
| ≥2 | 19(23.5) | 14(5.8) | | | |

@antepartum, intrapartum or postpartum hemorrhage, prolonged or obstructed labor, Complication of abortion, Hypertensive disorders of pregnancy

**Table 3. Clinical characteristics of maternal near misses of mothers admitted in tertiary hospitals, Southern Ethiopia, 2020 (n = 81).**

| Maternal near-miss events | Frequency (%) |
|---|---|
| **Potentially life-threatening conditions** | 71(87.6) |
| Severe postpartum hemorrhage | 41(50.6) |
| Severe preeclampsia | 11(13.6) |
| Eclampsia | 7(8.6) |
| Sepsis or severe systemic infection | 19(23.4) |
| Ruptured uterus | 12(14.8) |
| Severe anemia | 9(11.1) |
| **Critical interventions** | 39(48.1) |
| Use of blood products (includes any blood transfusion) | 21(25.9) |
| Interventional radiology (uterine artery embolization) | 0(0.0) |
| Laparotomy | 10(12.3) |
| Admission to Intensive Care Unit (ICU) | 13(16.0) |
| **Organ dysfunction / life-threatening conditions** | 5(6.2) |
| Cardiovascular dysfunction (shock, the sudden absence of pulse and loss of consciousness, use of continuous vasoactive drugs) | 1(1.2) |
| Respiratory dysfunction(acute cyanosis, severe tachypnea (respiratory rate>40 bpm), severe bradypnea (respiratory rate<6 bpm), or intubation and ventilation not related to anesthesia) | 2(2.5) |
| Renal dysfunction(oliguria non responsive to fluids or diuretics, or severe acute azotemia (creatinine 3.5mg/dL) | 1(1.2) |
| Hepatic dysfunction (jaundice, or severe acute hyperbilirubinemia (bilirubin >6.0mg/dL) | 0(0.0) |
| Neurological dysfunction (prolonged unconsciousness / coma (lasting >12 hours) | 0(0.0) |
| Uterine dysfunction (hemorrhage or infection leading to hysterectomy) | 2(2.5) |
| Coagulation/hematological dysfunction (failure to form clots, or massive transfusion of blood or red cells (5 units) or severe acute thrombocytopenia (<50,000 platelets/ml) | 0(0.0) |

## Medical conditions among respondents

More than half (65.4%) of the cases and one-third (33.2%) of the controls had a history of at least one pre-existing medical problem. Among the cases, anemia was the most common condition (25.9%), followed by hypertension (23.4%), diabetes mellitus (19.7%), and syphilis (6.2%). On the other hand, hypertension, diabetes mellitus, anemia, and syphilis were reported in 13.3%, 7.9%, 6.6%and 5.4%of controls.

## Clinical characteristics of maternal near misses

By using the WHO near-miss criteria for maternal health, near-miss cases were identified. Among potentially life-threatening conditions, severe postpartum hemorrhage and sepsis were the commonest reasons for case admission, 50.6%, and 23.4%, respectively. Twenty-one of them received a blood transfusion (25.9%). Just five(6.2%) of near-miss cases have been diagnosed with at least one type of organ dysfunction (Table 3).

## Maternal health service-related characteristics

Only 13.6% of cases and 19.5% of controls had an early booking for ANC follow-up and the proportion of cases with no ANC was double that of controls. Regarding the place delivery, nearly one-fifth (19.8%) of cases and 25(10.4%) of controls gave birth at home. The average time to decide to seek health care among the cases and controls was 36.0 and 24.3 hours, respectively. The predominant reasons for the first delay among controls were a lack of money

**Table 4. Maternal health service-related characteristics of mothers admitted to tertiary hospitals in Southern Ethiopia, 2020.**

| Variable categories | Cases = 81 | Controls = 221 | Total = 322 | $X^2$ | P-value |
|---|---|---|---|---|---|
| | n(%) | n(%) | n (%) | | |
| **ANC visit** | | | | | |
| > = 4 | 14(17.3) | 74(30.7) | 88(27.4) | 17.475 | 0.001 |
| 2–3 | 19(23.5) | 77(32.0) | 96(29.8) | | |
| 1 | 21(25.9) | 56(23.2) | 77(23.9) | | |
| No | 27(33.3) | 34(14.1) | 61(18.9) | | |
| **ANC booking** | | | | | |
| Early booking(<12 week) | 11(13.6) | 47(19.5) | 58(18.1) | 1.440 | 0.230 |
| Late booking(≥12week) | 70(86.4) | 194(80.5) | 264(81.9) | | |
| **Place of last delivery** | | | | | |
| Hospital | 47(58.0) | 178(73.9) | 225(69.9) | 7.816 | 0.020 |
| Health center | 18(22.2) | 38(15.9) | 56(17.4) | | |
| Home | 16(19.8) | 25(10.4) | 41(12.7) | | |
| **Mode of delivery** | | | | | |
| SVD | 46(56.8) | 113(46.9) | 159(49.4) | 2.598 | 0.273 |
| Instrumental delivery | 24(29.6) | 82(34.0) | 106(32.9) | | |
| C/S | 11(13.6) | 46(19.1) | 57(17.7) | | |
| **Induction of labor** | | | | | |
| Yes | 23(28.4) | 92(38.2) | 115(35.7) | 2.592 | 0.072 |
| No | 58(71.6) | 149(61.8) | 207(64.3) | | |
| **Knowledge of danger signs** | | | | | |
| Yes | 49(60.5) | 179(74.3) | 228(70.8) | 5.569 | 0.024 |
| No | 32(39.5) | 62(25.7) | 94(29.2) | | |
| **Means of transportation** | | | | | |
| On foot | 36(44.4) | 117(48.5) | 153(47.5) | | |
| Rented transport | 25(30.9) | 73(30.3) | 98(30.4) | | |
| Ambulance | 20(24.7) | 51(21.2) | 71(22.1) | | |
| **Autonomy in decision making** | | | | | |
| Yes | 32(39.5) | 119(49.4) | 151(46.9) | 2.372 | 0.157 |
| No | 49(60.5) | 122(50.6) | 171(53.1) | | |
| **First Delay** | | | | | |
| Yes (>24hr) | 45(55.6) | 118(49.0) | 163(50.6) | 1.054 | 0.305 |
| No (≤24hr) | 36(44.4) | 123(51.0) | 159(49.4) | | |
| **Second delay** | | | | | |
| Yes (>60min) | 43(53.1) | 53(22.0) | 96(29.8) | 28.012 | <0.001 |
| No (≤60min) | 38(46.9) | 188(78.0) | 226(70.2) | | |
| **Third delay** | | | | | |
| Yes(>60 min) | 33(40.7) | 71(29.5) | 104(32.3) | 3.52 | 0.060 |
| No(≤60min) | 48(59.3) | 170(70.5) | 218(67.7) | | |

and not considering the symptoms as severe. The lack of transportation and the failure to consider the symptoms as severe were the main reasons for the first delay among the cases. In terms of the second delay, the median time spent walking to the nearest health facility among the cases and controls was 40.3 and 61.2 minutes, respectively (Table 4).

**Birth preparedness and complication readiness (BPCR) plan.** Just 14.8% of cases and 42.3% of controls had good practice of BPCR. As for the percentages of individual BPCR components, more than six out of ten cases (60.5%) and nearly three-fourths (73.4%) of controls

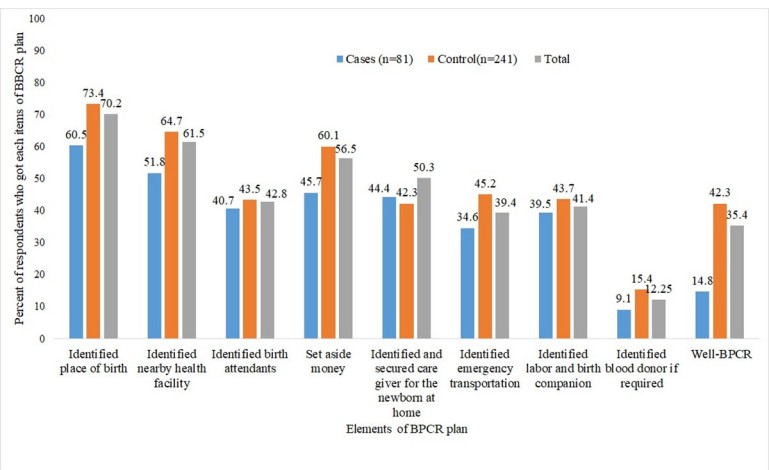

**Fig 1. Shows the percentages of BPCR practice of women admitted to tertiary hospitals in Southern Ethiopia, 2020.**

identified the place of birth, while only 11.1% of cases and 15.4% of controls identified blood donors if required (Fig 1).

## Determinants of maternal near-miss (MNM)

Multivariable logistic regression analysis showed that lack of ANC, previous history of cesarean section, delay in accessing the final place of treatment for more than 60 minutes, poor BPCR practice, and history of pre-existing medical disorders were significantly associated with MNM.

The lack of antenatal care was proved to be a major determinant of the MNM. Compared to women who received four or more prenatal visits, women who did not receive ANC had a 3.2 times greater risk of having maternal near-miss [AOR = 3.25; 95% CI: 2.21, 7.69]. Similarly, compared to women with no prior history of cesarean section, women with the previous history of Cesarean section had 3.5 times higher chances of developing maternal near-miss [AOR = 3.53; 95% CI: 1.79, 6.98]. In the current study, one of the factors positively associated with MNM is birth preparedness and complication readiness. In women with poor birth preparedness and complication readiness (BPCR) plan, the risk of maternal near was 3.3 times higher compared to a well-prepared one [AOR = 3.31; 95% CI: 1.50, 7.29].

Moreover, women with at least one pre-existing medical condition were 2.8 times more likely to experience a maternal near-miss than their counterparts [OR = 2.79; 95% CI:1.45, 5.37]. Compared to their counterparts, women who traveled more than 60 minutes to reach their final place of care had 3.2 times higher chances of experiencing MNM[AOR = 3.21; 95 percent CI:1.61, 6.39] (Table 5).

## Discussion

The lack of ANC, previous history of cesarean section, delaying more than 60 minutes to reach the final place of treatment, the poor practice of BPCR, and history of pre-existing medical disorders have been identified in the current study as determinants of a maternal near-miss.

In this study, the lack of ANC visits was positively related to maternal near-miss and is tandem with studies conducted in Nigeria, Brazil, Iraq, Morocco, and Nigeria, which support the receipt of adequate ANC as a shielding factor against severe maternal outcomes and near-miss

**Table 5. Determinants of MNM among women admitted to tertiary hospitals in Southern Ethiopia, 2020.**

| Variable Categories | MNM | | COR(95%CI) | AOR(95%CI) | p-value |
|---|---|---|---|---|---|
| | Cases (%) | Controls (%) | | | |
| **Age** | | | | | |
| 35+ | 10(12.2) | 22(9.1) | 1 | 1 | |
| 20–34 | 63(77.9) | 206(85.5) | 0.67(0.30,1.49) | 0.58(0.19,1.76) | 0.336 |
| <20 | 8(9.9) | 13(5.4) | 1.35(0.42,4.29) | 1.03(0.18,3.83) | 0.975 |
| **Residence** | | | | | |
| Urban | 40(49.4) | 148(61.4) | 1 | 1 | |
| Rural | 41(50.6) | 93(38.6) | 1.63(0.98,2.71)* | 1.22(0.65,2.30) | 0.49 |
| **Mother's Educational level** | | | | | |
| College and above | 10(12.3) | 44(18.2) | 1 | 1 | |
| Secondary(9-12th) | 13(16.0) | 63(26.1) | 0.91(0.36,2.25) | 0.91(0.30,2.75) | 0.869 |
| Primary education (1-8th) | 16(19.8) | 52(21.6) | 1.35(0.56,3.28) | 1.37(0.46,4.10) | 0.570 |
| No formal education | 42(51.9) | 82(34.0) | 2.25(1.03,4.92) | 1.41(0.53,3.74) | 0.494 |
| **Wealth index** | | | | | |
| Highest | 13(16.0) | 51(21.2) | 1 | 1 | |
| Fourth | 13(16.0) | 52(21.6) | 0.98(0.41,2.32) | 1.04(0.35,2.86) | 0.995 |
| Middle | 16(19.8) | 48(19.9) | 1.31(0.57,3.00) | 1.82(0.63,4.26) | 0.265 |
| Second | 19(23.5) | 46(19.1) | 1.62(0.72,3.64) | 1.33(0.46,3.85) | 0.603 |
| Lowest | 20(24.7) | 44(18.3) | 1.78(0.79,3.99) | 2.42(0.86,5.76) | 0.092 |
| **Age at first pregnancy** | | | | | |
| ≥20 | 40(49.1) | 166(68.9) | 1 | 1 | |
| 17–19 | 25(30.9) | 50(20.7) | 2.07(1.15,3.75) | 1.38(0.64,2.99) | 0.413 |
| <16 | 16(19.8) | 25(10.4) | 2.65(1.29,5.44)* | 1.41(0.54,3.67) | 0.485 |
| **Birth interval** | | | | | |
| ≥24 months | 38(46.9) | 163(67.6) | 1 | 1 | |
| <24 months | 43(53.1) | 78(32.4) | 2.36(1.41,3.95)* | 1.58(0.83,3.02) | 0.162 |
| **Desire on the last pregnancy** | | | | | |
| Planned | 48(59.3) | 169(70.1) | 1 | 1 | |
| Not planned | 33(40.7) | 72(29.9) | 1.61(0.96,2.72)* | 1.44(0.73,2.84) | 0.291 |
| **Previous history of C/S** | | | | | |
| No | 41(50.6) | 195(80.9) | 1 | 1 | |
| Yes | 40(49.4) | 46(19.1) | 4.14(2.41,7.11)* | **3.53(1.79,6.98)**** | <0.001 |
| **ANC visit** | | | | | |
| > = 4 | 14(17.3) | 74(30.7) | 1 | 1 | |
| 2–3 | 19(23.5) | 77(32.0) | 1.30(0.61,2.79) | 1.23(0.48,3.19) | 0.646 |
| 1 | 21(25.9) | 56(23.2) | 1.98(0.93,4.24) | 1.37(0.52,3.60) | 0.524 |
| No | 27(33.3) | 34(14.1) | 4.19(1.96,8.99)* | **3.25(2.21,7.69)**** | 0.019 |
| **ANC booking** | | | | | |
| Early booking(<12 week) | 11(13.6) | 47(19.5) | 1 | | |
| Late booking(≥12week) | 70(86.4) | 194(80.5) | 1.54(0.76,3.14) | 1.24(0.83, 2.12) | 0.231 |
| **Place of last delivery** | | | | | |
| Hospital | 47(58.0) | 178(73.9) | 1 | | |
| Health center | 18(22.2) | 38(15.9) | 1.79(0.94,3.42) | 1.34(0.53,3.43) | 0.519 |
| Home | 16(19.8) | 25(10.4) | 2.42(1.19,4.91)* | 1.62(0.67,3.90) | 0.184 |
| **BPCR plan** | | | | | |
| Good | 12(14.8) | 102(42.3) | 1 | 1 | |
| Poor | 69(85.2) | 139(57.7) | 4.22(2.17,7.19)* | **3.31(1.50,7.29)**** | 0.003 |

(*Continued*)

**Table 5.** (Continued)

| Variable Categories | MNM | | COR(95%CI) | AOR(95%CI) | p-value |
|---|---|---|---|---|---|
| | Cases (%) | Controls (%) | | | |
| **Knowledge of danger signs during pregnancy** | | | | | |
| Yes | 49(60.5) | 179(74.3) | 1 | 1 | |
| No | 32(39.5) | 62(25.7) | 1.88(1.11,3.20)* | 1.06(0.52,2.15) | 0.873 |
| **Autonomy in decision making** | | | | | |
| Yes | 32(39.5) | 119(49.4) | 1 | | |
| No | 49(60.5) | 122(50.6) | 1.49(0.89,2.49)* | 1.61(0.82,3.16) | 0.164 |
| **Second delay** | | | | | |
| No(≤60min) | 38(46.9) | 188(78.0) | 1 | 1 | |
| Yes(>60 min) | 43(53.1) | 53(22.0) | 4.01(2.36,6.83)* | **3.21(1.61,6.39)**** | 0.001 |
| **Third delay** | | | | | |
| No(≤60min) | 48(59.3) | 170(70.5) | 1 | 1 | |
| Yes(>60 min) | 33(40.7) | 71(29.5) | 1.65(0.97,2.77)* | 1.62(0.82,3.23) | 0.167 |
| **Preexisting chronic medical conditions** | | | | | |
| No | 28(34.6) | 161(66.8) | 1 | 1 | |
| Yes | 53(65.4) | 80(33.2) | 3.81(2.24,6.47)* | **2.79(1.45,5.37)**** | 0.002 |

**Key:** 1: Reference category; AOR = Adjusted odds ratio, COR = Crude odds ratio

*statistically significant at p-value<0.25

** statistically significant at p-value <0.05

[12, 39–41]. Besides, research in Ethiopia found that women who did not have antenatal visits were more likely to experience maternal near-misses [11, 42, 43]. The potential justification may be antenatal care is the most important touch point for mothers to get more information about danger signs of pregnancy and childbirth by consultation with health professionals. If a mother lacks ANC, minor obstetric conditions are not detected and managed early, serious complications and MNM will likely develop. This finding is contradictory to studies carried out in Bolivia [44] and northern Ethiopia [45], which indicated that routine ANC has an indirect impact on a maternal near-miss, likely by preventing the delay in seeking treatment through raising awareness of timely care. Therefore, the concerted effort needed by health care providers to track those mothers without antenatal care is key in reducing the incidence of MNM.

In line with previous studies conducted in Northeast Brazil, the Netherlands, South Africa, and Ethiopia [16, 40, 46–48], the current research also found that women with a previous history of cesarean section were more prone to the maternal near-miss event. A global maternal near-miss survey also indicated that previous cesarean section experience was a risk factor for maternal near-misses [49]. Despite its benefit in saving the life of a woman and newborn, halting pregnancy with a cesarean section raises the risk of infection, hemorrhage, thromboembolism, uterine scar, and uterine rupture, which may increase the probability of MNM [50, 51]. This result showed that the possible risk of the cesarean section should be taken into consideration by health care providers and it should be done in the presence of convincing clinical indications. In other words, to minimize health risks associated with cesarean section, non-medical indications of delivery by cesarean section should be deterred to the acceptable level recommended by WHO(5–15%) [52].

In the current study, the time taken to reach the final point of treatment (second delay) has been found to increase the likelihood of MNM and is strengthened by studies conducted in

Nigeria, Sierra Leone, Morocco, and Northern Ethiopia [12, 16, 39, 53]. Poor access to local health facilities, a lack of road infrastructure or transportation causes women to travel long distances on foot, resulting in delays in accessing health facilities, and this might lead to a near-miss event [54]. We suggest that by improving access to roads and other transport facilities such as emergency services, regional and zonal stakeholders must increase their efforts to reduce maternal near-miss occurrences. Besides, access to health facilities needs to be revised by decentralizing maternity care through the establishment of comprehensive emergency obstetric care (CEMOC) centers.

Our research showed that among mothers with a history of a pre-existing chronic medical condition, the odds of maternal near-miss were higher. Several studies in countries such as Nigeria, Ghana, Sudan, and Ethiopia have also documented that, the history of anemia and chronic hypertension resulted in a maternal near miss [12, 16, 18, 55]. The likelihood of complications such as superimposed pre-Eclampsia, placental abruption, intrauterine growth retardation, and pre-term delivery is substantially enhanced by those chronic conditions, all of which may be triggers for referral to higher facilities [8, 56]. To minimize MNM, promoting early screening and treatment of non-communicable diseases by health care providers should also be a good measure.

In the current study, one of the factors positively associated with MNM is the practice of birth preparedness and complication readiness. Women who were not well prepared for birth and their complications were more likely to encounter MNM events. This may be because women with a poor BPCR plan were likely to be exposed to one of the three delays (such as delays in seeking, reaching, and receiving care) and thus in favour of MNM events [57]. This result is a new addition to this study and has a policy implication as BPCR is one of WHO's twelve key recommendations to enhance the use of skilled maternity care and to reduce serious obstetric conditions through the well-timed use of facility care [22]. This finding calls for more attention to the improvements in practice for women on BPCR.

## Strength and limitation of the study

The current study conducted on highly equipped hospitals and focused on all the three WHO criteria, unlike other studies that concentrate only on potentially life-threatening conditions and ignore the clinical criteria (i.e. organ dysfunction and provision of clinical interventions). This may make the evaluation criteria more stringent and show a better picture of near-miss events. Although senior doctors working in the study hospitals verified the reported cases, there may be misclassification bias. Because the nature of the study was Unmatched case-control, confounders are hard to control because cases and controls are not matched with relevant variables. Since the study was based on self-reports, the respondents might be prone to social desirability bias. Finally, because women were asking about incidents that had already occurred during the last year before this study, there may be a risk of recall bias.

## Conclusion

In the current study, the lack of ANC, prior history of Cesarean section, delaying more than 60 minutes to reach the final place of care, poor BPCR practice, and history of pre-existing medical conditions were established as determinants of a maternal near-miss. Stakeholders at the zonal and regional levels need to enhance their efforts for improving access to roads and transportations facilities like ambulance services. Besides, women who have a prior history of cesarean section, chronic medical conditions, and no ANC need special attention from their families and health care providers to proactively mitigate the occurrence of serious obstetric

complications. More attention has to be paid to birth preparedness and complication readiness activities.

## Supporting information

**S1 Data. Data collection tool used to identify determinants of maternal near miss.**
(DOCX)

**S2 Data. The raw data supporting the findings of this article.**
(SAV)

## Acknowledgments

We are indebted to Wachemo University College of medicine and health science, department of public health for giving Ethical clearance to undertake the study. Our appreciation also goes to the managers and healthcare providers who worked in the selected hospitals for their assistance and cooperation during the study. Finally, for their efforts, we want to thank our supervisors, data collectors, and study participants.

## Author Contributions

**Conceptualization:** Aklilu Habte.

**Data curation:** Aklilu Habte.

**Formal analysis:** Aklilu Habte.

**Investigation:** Aklilu Habte, Merertu Wondimu.

**Methodology:** Aklilu Habte.

**Project administration:** Aklilu Habte.

**Resources:** Aklilu Habte, Merertu Wondimu.

**Supervision:** Aklilu Habte, Merertu Wondimu.

**Validation:** Aklilu Habte.

**Visualization:** Aklilu Habte.

**Writing – original draft:** Aklilu Habte.

**Writing – review & editing:** Aklilu Habte, Merertu Wondimu.

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
