## [Decision Letter · Decision Letter 0]

10 Mar 2021

PONE-D-20-40965

Determinants of Maternal Near Miss among Women Admitted to Maternity Wards of Tertiary Hospital in Southern Ethiopia, 2020: A Hospital-Based Case-control Study

PLOS ONE

Dear Dr. Hailegebireal,

Thank you for submitting your manuscript to PLOS ONE. After careful consideration, we feel that it has merit but does not fully meet PLOS ONE’s publication criteria as it currently stands. Therefore, we invite you to submit a revised version of the manuscript that addresses the points raised during the review process.

We look forward to receiving your revised manuscript.

Kind regards,

Ricardo Q. Gurgel, PhD

Academic Editor

PLOS ONE

Journal Requirements:

2. Please include additional information regarding the survey or questionnaire used in the study and ensure that you have provided sufficient details that others could replicate the analyses. For instance, if you developed a questionnaire as part of this study and it is not under a copyright more restrictive than CC-BY, please include a copy, in both the original language as well as the English version already provided, as Supporting Information.

3. In the Methods, please clarify that participants provided oral consent. Please also state in the Methods:

- Why written consent could not be obtained

- Whether the Institutional Review Board (IRB) approved use of oral consent

- How oral consent was documented

For more information, please see our guidelines for human subjects research: https://journals.plos.org/plosone/s/submission-guidelines#loc-human-subjects-research

4. You indicated that you had ethical approval for your study. In your Methods section, please ensure you have also stated whether you obtained consent from parents or guardians of the minors included in the study or whether the research ethics committee or IRB specifically waived the need for their consent.

5. Please ensure you have thoroughly discussed any potential limitations of this study within the Discussion section, including the potential impact of confounding factors.

Reviewers' comments:

Reviewer's Responses to Questions

**Comments to the Author**

1. Is the manuscript technically sound, and do the data support the conclusions?

Reviewer #1: Yes

Reviewer #2: Yes

2. Has the statistical analysis been performed appropriately and rigorously? 

Reviewer #1: Yes

Reviewer #2: Yes

3. Have the authors made all data underlying the findings in their manuscript fully available?

Reviewer #1: Yes

Reviewer #2: Yes

4. Is the manuscript presented in an intelligible fashion and written in standard English?

Reviewer #1: Yes

Reviewer #2: Yes

5. Review Comments to the Author

Reviewer #1: Overall information

The authors have studied one of the most important areas in maternal health research. Studies like these are really important to assess the prevalence, trend and determinant factors of maternal near-misses across different setting. Methodologically the study seems somehow good and the conclusions were almost valid on the bases of the data presented. The data were presented clearly and appropriately. However, I can suggest some major and minor revision on the paper.

Major issues

1. Under introduction section it was stated that, in the current study the authors have used all the three WHO criteria to make the evaluation of maternal near miss more stringent. However under data collection tool, procedure and personnel, the authors stated that they have used WHO disease specific criteria. [Page 2, line 19-21 and page 6, line 4]

2. Cases were selected during pregnancy, delivery or postpartum period while controls were selected during pregnancy or post-partum period. The authors did not selected controls during delivery. Hence, controls were not comparable and selection bias is likely. [Page 5, Line 2-4 and 10-12]

4. The authors did not explain how the wealth index was assessed and analyzed. However, it has been reported under result part. [Page 9, Table 1]

Minor issues

1. The authors indicated that they have used 1:3 ratios for cases and controls. However, the final calculated sample was not in exactly 1:3 ratio. [Page 5, line 21 and 26]

2. The authors considered different hospital; this was good to avoid factors that could be unique to a given hospital as a result of the referral pattern. However, the authors did not indicate what the selected hospitals were. What the pattern of cases and controls from each hospital look like. [Page 5, line 28].

3. The authors stated that controls were picked using systematic methods of sampling. This needs to be elaborated clearly. [Page5, line 32]

4. The authors need to clearly explain the source population for cases and controls

5. The authors stated the have used standard tool after possible adjustment to the local context. What adjustment was actually made? [Page 6, line 4]

6. Under result section I do not see the advantage of repetition of variables in table 2 and table 3

7. Remove figure 1 title from page 13.

8. The resolution for figure 1 need improvement.

9. In order to maintain the logical flow of results, I suggest the authors shall put maternal health service-related characteristics immediately before determinant of maternal near-miss

10. Under discussion part in the last paragraph the authors stated the result could not be inferred to the general population. Can they clearly specify and elaborate this statement?

11. Under conclusion section the authors have recommended, stakeholders at the zonal and regional levels need to enhance their efforts for improving access to roads and transportations facilities like ambulance services. I could not see any data from the study supporting this recommendation. The study did not assessed infrastructures and ambulance service. [Page19, line 31-32]

Miscellaneous remarks

1. Typing error, eg. Cases were not 221 rather 241 in tables under result section. The authors shall edit for some grammatical and punctuation .issues. Besides, unnecessary bold letters, words and numbers should be revised from the entire document especially under reference.

2. The authors did not acknowledged the strength and limitation of the study

3. Reference should be improved in such a way that it sticks to the standard. Eg. reference 16 and 31 were super scripted. [Page 4, line 17]

Reviewer #2: Thank you for giving me this chance to review an interesting area in maternal health

General query

The topic has been discussed many times in the Ethiopian context, what new information do you think you brought to the scientific world from your current study?

The manuscript has no line numbers, which made the review and putting comments and questions quit challenging

Abstract

How would you justify this sentence in the abstract section “Ethiopia is one of the countries in sub-Saharan Africa with high maternal mortality and morbidity, but there is little evidence of risk factors for a near miss”? Because I have seen different studies conducted on the issue in Ethiopia

Introduction

The same question as the above one goes to the first sentence in paragraph 6 of the introduction section, “Ethiopia is one of the countries in sub-Saharan Africa with high maternal mortality and

morbidity, but there is little evidence of risk factors for a near miss.”

Method

On the study setting, would you describe the 5 selected hospitals of the study?

Result

On subsection maternal health service-related characteristics, line 2, re-write as Regarding the place of birth

Same section line 5: capitalize the letter t, The predominant reasons.....

Discussion

Paragraph 3 ,line 9: in the sentence..... presence of convincing clinical signals. Do you mean signs?

The last paragraph on the discussion speaks about strength and limitation, so provide the appropriate heading for the paragraph

6. PLOS authors have the option to publish the peer review history of their article (what does this mean?). If published, this will include your full peer review and any attached files.

Reviewer #1: No

Reviewer #2: No

---

## [Author Response · Author response to Decision Letter 0]

22 Mar 2021

The response to reviews have been attached as a "Response to Reviewers" in the submission process

---

## [Decision Letter · Decision Letter 1]

26 Apr 2021

PONE-D-20-40965R1

Determinants of Maternal Near Miss among Women Admitted to Maternity Wards of Tertiary Hospitals in Southern Ethiopia, 2020: A Hospital-Based Case-control Study

PLOS ONE

Dear Dr. Hailegebireal,

Thank you for submitting your manuscript to PLOS ONE. After careful consideration, we feel that it has merit but does not fully meet PLOS ONE’s publication criteria as it currently stands. Therefore, we invite you to submit a revised version of the manuscript that addresses the points raised during the review process.

Please make the few remainning changes for the final acceptance.

We look forward to receiving your revised manuscript.

Kind regards,

Ricardo Q. Gurgel, PhD

Academic Editor

PLOS ONE

Journal Requirements:

Reviewers' comments:

Reviewer's Responses to Questions

**Comments to the Author**

1. If the authors have adequately addressed your comments raised in a previous round of review and you feel that this manuscript is now acceptable for publication, you may indicate that here to bypass the “Comments to the Author” section, enter your conflict of interest statement in the “Confidential to Editor” section, and submit your "Accept" recommendation.

Reviewer #1: All comments have been addressed

Reviewer #2: (No Response)

2. Is the manuscript technically sound, and do the data support the conclusions?

Reviewer #1: Yes

Reviewer #2: Yes

3. Has the statistical analysis been performed appropriately and rigorously? 

Reviewer #1: Yes

Reviewer #2: Yes

4. Have the authors made all data underlying the findings in their manuscript fully available?

Reviewer #1: Yes

Reviewer #2: Yes

5. Is the manuscript presented in an intelligible fashion and written in standard English?

Reviewer #1: Yes

Reviewer #2: Yes

6. Review Comments to the Author

Reviewer #1: I would like to thank the authors for their effort to consider my comment. I have thoroughly gone through the document and I found that all my comment were properly addressed. However, I am still having a concern regarding the resolution of Figure 1.

Reviewer #2: Thank you for addressing my previous comments. I am very impressed with the level of effort authors have made to revise their publication. But still there are some modifications I consider should be emphasized on

1. On the abstract section, modifying the sentence with “As a result, assessing and identifying MNM determinants will aid in improving the capacity of the health system to reduce severe maternal morbidity and mortality” will not address the concern I have raised previously. Your justification should address the importance or uniqueness of your study, removing the statement would take you back my primary concern , which was: How would you justify this sentence in the abstract section “Ethiopia is one of the countries in sub-Saharan Africa with high maternal mortality and morbidity, but there is little evidence of risk factors for a near miss”? I have seen different studies conducted on the issue in Ethiopia

2. The line numbers should be continuous (on the manuscript you revised), and I belive while referring the points you addressed, you should mention the lines numbers on the revised manuscript

3. The issue I raised regarding the description of the five hospitals should be on the study area, in addition not only mentioning the name of the hospitals but also describing the hospitals in relation to your study objective (in the terms of number of obstetrics cases treated annually, facility related characteristics like the readiness of the facilities in managing cases of MNM....)

7. PLOS authors have the option to publish the peer review history of their article (what does this mean?). If published, this will include your full peer review and any attached files.

Reviewer #1: **Yes: **Firew Tiyare Tiruneh

Reviewer #2: No

---

## [Author Response · Author response to Decision Letter 1]

27 Apr 2021

The response to reviews have been attached as a "Response to Reviewers" in the submission process

---

## [Editor Report · Decision Letter 2]

4 May 2021

Determinants of Maternal Near Miss among Women Admitted to Maternity Wards of Tertiary Hospitals in Southern Ethiopia, 2020: A Hospital-Based Case-control Study

PONE-D-20-40965R2

Dear Dr. Habte,

We’re pleased to inform you that your manuscript has been judged scientifically suitable for publication and will be formally accepted for publication once it meets all outstanding technical requirements.

Kind regards,

Ricardo Q. Gurgel, PhD

Academic Editor

PLOS ONE
---

## [Editor Report · Acceptance letter]

6 May 2021

PONE-D-20-40965R2 

Determinants of Maternal Near Miss among Women Admitted to Maternity Wards of Tertiary Hospitals in Southern Ethiopia, 2020: A Hospital-Based Case-control Study 

Dear Dr. Habte:

I'm pleased to inform you that your manuscript has been deemed suitable for publication in PLOS ONE. Congratulations! Your manuscript is now with our production department. 

Kind regards, 

on behalf of

Professor Ricardo Q. Gurgel 

Academic Editor

PLOS ONE